# Glycosphingolipids with Very Long-Chain Fatty Acids Accumulate in Fibroblasts from Adrenoleukodystrophy Patients

**DOI:** 10.3390/ijms22168645

**Published:** 2021-08-11

**Authors:** Yuko Fujiwara, Kotaro Hama, Nobuyuki Shimozawa, Kazuaki Yokoyama

**Affiliations:** 1Faculty of Pharma-Science, Teikyo University, Itabashi-ku, Tokyo 173-8605, Japan; khama@pharm.teikyo-u.ac.jp (K.H.); yokoyama@pharm.teikyo-u.ac.jp (K.Y.); 2Division of Genomics Research, Life Science Research Center, Gifu University, Gifu 501-1193, Japan; nshim@gifu-u.ac.jp

**Keywords:** adrenoleukodystrophy, glycosphingolipid, very-long-chain fatty acids

## Abstract

Adrenoleukodystrophy (X-ALD) is an X-linked genetic disorder caused by mutation of the ATP-binding cassette subfamily D member 1 gene, which encodes the peroxisomal membrane protein, adrenoleukodystrophy protein (ALDP). ALDP is associated with the transport of very-long-chain fatty acids (VLCFAs; carbon chain length ≥ 24) into peroxisomes. Defective ALDP leads to the accumulation of saturated VLCFAs in plasma and tissues, which results in damage to myelin and the adrenal glands. Here, we profiled the glycosphingolipid (GSL) species in fibroblasts from X-ALD patients. Quantitative analysis was performed using liquid chromatography–electrospray ionization–tandem mass spectrometry with a chiral column in multiple reaction monitoring (MRM) mode. MRM transitions were designed to scan for precursor ions of long-chain bases to detect GSLs, neutral loss of hexose to detect hexosylceramide (HexCer), and precursor ions of phosphorylcholine to detect sphingomyelin (SM). Our results reveal that levels of C25 and C26-containing HexCer, Hex2Cer, NeuAc-Hex2Cer, NeuAc-HexNAc-Hex2Cer, Hex3Cer, HexNAc-Hex3Cer, and SM were elevated in fibroblasts from X-ALD patients. In conclusion, we precisely quantified SM and various GSLs in fibroblasts from X-ALD patients and determined structural information of the elevated VLCFA-containing GSLs.

## 1. Introduction

X-linked adrenoleukodystrophy (X-ALD) is an inherited neurodegenerative disease and the most common peroxisomal disorder. X-ALD is caused by mutation of the ATP-binding cassette subfamily D member 1 (*ABCD1*) gene, whose encoded protein, adrenoleukodystrophy protein (ALDP), is involved in the transport of very-long-chain fatty acyl-CoA esters across the peroxisomal membrane. Thus, mutation in the *ABCD1* gene results in the accumulation of very-long-chain fatty acids (VLCFAs) in plasma, fibroblasts, and tissues. In this study, fatty acids (FAs) with 24 or more carbon atoms are described as VLCFAs. The clinical presentation of X-ALD patients is diverse, with at least six distinguishable clinical phenotypes. The two main clinical phenotypes are childhood cerebral ALD (CCALD) and adrenomyeloneuropathy (AMN) [1]. CCALD is the most rapidly progressing phenotype that mainly occurs in boys between the ages of 3 and 10 and is characterized by severe inflammatory demyelination in the brain, leading to progressive intellectual, psychiatric, visual, and gait disturbances [2,3]. In contrast, AMN has a less severe clinical manifestation [4] and is characterized by a slowly progressive distal axonopathy. The first symptoms of AMN usually appear in male patients in the second to fourth decade of life [5]. A newborn male with X-ALD is likely to develop AMN in adulthood. In heterozygous women, the age at onset of AMN is the fourth to fifth decade of life. Mean motor disability and progression is less severe than in males. However, heterozygous women develop more severe gait disturbance than males because of involvement of spinal cord sensory pathways [6]. The distribution/incidence of each phenotype varies from country to country [7,8]. The clinical phenotypes of X-ALD can be highly variable, and the phenotype cannot be predicted by the nature of the *ABCD1* pathogenic variant, as the same pathogenic variant can be associated with each of the known phenotypes. Mild phenotypes may be associated with large deletions that abolish formation of the gene product, and severe phenotypes occur with missense pathogenic variants in which abundant immunoreactive protein product is produced [9,10,11,12]. Therefore, the various clinical types of X-ALD can be observed in the same kindreds and nuclear families carrying the same mutation in the *ABCD1* gene [13,14,15], and genotype–phenotype correlation in X-ALD has not been demonstrated.

Measurement of plasma VLCFA levels, particularly C26:0 VLCFA, has been the standard test for X-ALD diagnosis; however, most fatty acyl chains are not present in their free form in biological samples but are incorporated into complex lipids such as phospholipids (PLs) [16]. Recent advances in comprehensive lipidomic analysis by liquid chromatography–tandem mass spectrometry (LC–MS/MS) have revealed the endogenous form of VLCFAs in X-ALD patients to comprise PLs, glycolipids, and neutral lipids [17,18,19]. Although increases in levels of VLCFAs, cholesteryl esters, cerebrosides, sulfatides, and gangliosides have been detected in X-ALD patient brains [20,21,22,23,24,25,26], comprehensive lipidomic analyses of glycosphingolipid species by LC–MS have not been conducted to date.

Here, we profiled glycosphingolipid (GSL) species in fibroblasts from X-ALD patients to identify altered GSL species and to elucidate the mechanism of disease onset.

## 2. Results

### 2.1. Quantitative Analysis of GSLs in Fibroblasts from X-ALD Patients

GSLs are synthesized by the sequential addition of carbohydrate and carbohydrate derivatives to ceramide. Our LC–MS method was unable to distinguish between galactose and glucose or to determine the linkage of constituent sugars, such as N-acetylhexosamine and N-acetylneuraminic acid. Therefore, we named the GSLs detected in our study according to results using standard materials and with reference to previous studies of human skin fibroblasts. A summary of the abbreviated name, systematic name, common name, and relevant references for each GSL is provided in Table 1. The structures of the lipids used for this study are listed in Figure 1.

We first profiled each GSL species in fibroblasts from X-ALD patients comprising four CCALD and three AMN patients and six control subjects using our recently developed LC–MS method that uses a chiral column in positive ion mode. Among the 54 molecular species (fatty acids with 14–30 carbons and 0–2 double bonds) for each GSL, 15 species of dihexosylceramide (Hex2Cer; LacCer), 19 species of GM3, 8 species of GM2, 4 species of GM1, 3 species of GD3, 18 species of Gb3, and 14 species of Gb4 were detected in fibroblasts from both X-ALD and control patients (Figure 2, Figure 3, Figure 4, Figure 5, Figure 6, Figure 7 and Figure 8). The amounts of C25:0-, C26:0-, and C26:1-containing LacCer were significantly higher in both CCALD and AMN groups compared with those in the control group (Figure 2). The amounts of C25:0-, C25:1-, C26:0-, and C26:1-containing GM3 were significantly higher in both CCALD and AMN groups compared with those in the control group, whereas the C26:2 species was significantly increased only in the CCALD group compared with that in the control group (Figure 3). In addition, the amounts of C25:0-, C26:0-, C26:1-, and C26:2-containing Gb3 were significantly higher in both CCALD and AMN groups compared with those in the control group, whereas the C25:1 species was significantly increased in the CCALD group but not in the AMN group compared with that in the control group (Figure 7). For GM2, only the amount of the C26:1 species was increased (Figure 4), but no species were significantly increased for GM1 or GD3 (Figure 5 and Figure 6). The amounts of C26:0- and C26:1-containing Gb4 were significantly higher in the CCALD group compared with those in the control group, whereas, in the AMN group, only C26:0-containing Gb4 was increased compared with that in the control group (Figure 8). No significant differences were observed in the amounts of any detected LacCer, GM3, GM2, GM1, GD3, Gb3, or Gb4 species between CCALD and AMN fibroblasts.

### 2.2. Quantitative Analysis of Hexosylceramide (HexCer) and Sphingomyelin (SM) in Fibroblasts from X-ALD Patients

Quantification of GSL profiles was based on precursor ion scans in positive ion mode at 264 *m*/*z*, which is indicative of long-chain base (LCB) ions. Although this method allowed comprehensive detection of various GSLs that contained d18:1, it was unable to separate HexCer and SM. This is because SM also contains d18:1 and generates a product ion at 264 *m*/*z* (Figure 9A,B). Moreover, the difference between the *m*/*z* values of the precursor ions and the retention time of specific SMs and HexCers is small. These features hamper the precise assignment of each lipid species. For example, GalCer d18:1/C18:0 was detected at 728.6 *m*/*z* with a retention time of 33.57 min, while SM d18:1/C18:1 was detected at 729.6 *m*/*z* with a retention time of 33.29 min (Figure 10). Structural analysis by MS/MS in positive ion mode at 730 *m*/*z* contained product ions generated from both GalCer d18:1/C18:0 and SM d18:1/C18:1 (Figure 9C,D). To overcome this issue, we applied a specific multiple reaction monitoring (MRM) transition setting to each HexCer and SM (Table 2). SM was detected in positive ion mode by a transition from the corresponding molecular ion to the *m*/*z* of 184, a characteristic ion of the phosphorylcholine headgroup (Figure 9B). Meanwhile, a neutral loss scan of 180 Da for the hexose residue in positive ion mode was applied to detect HexCer (Figure 9A). A total of 23 HexCer species were detected in both X-ALD and control fibroblasts, and HexCer 44:1, HexCer 44:2, and HexCer 44:3 were significantly more abundant in the CCALD group compared with those in the control group (Figure 11). Because quantification of HexCer and SM by MRM was not based on precursor ion scans in positive ion mode at 264 *m*/*z*, which is indicative of LCB ions, the combination of LCB and FA speices was not identified. Instead, only species that showed differences in amounts between X-ALD and control fibroblasts were analyzed by structural analysis. HexCer 44:1, HexCer 44:2, and HexCer 44:3 were identified as HexCer d18:1/26:0, HexCer d18:1/26:1, and HexCer d18:1/26:2, respectively, by electrospray ionization (ESI)-MS/MS. For SM, 27 species were detected in both X-ALD and control fibroblasts, and SM 44:1 and SM 44:2 were significantly more abundant in the CCALD group compared with those in the control group (Figure 12). SM 44:1 and SM 44:2 were identified as SM d18:1/26:0 and SM d18:1/26:1, respectively. Importantly, only one species, HexCer d18:1/C26:0, showed a significant difference in abundabnce between CCALD and AMN groups. The quantities of each GSL and SM species in fibroblasts from CCALD patients and comparisions of the quantities among CCALD, AMN, and control groups are shown in Figure 13.

## 3. Discussion

In this study, we detected four classes of ganglioside in human fibroblasts, GM3, GM2, GM1, and GD3. The most species were detected for GM3, followed by those for GM2, GM1, and GD3. Previously, Popa and Portoukalian reported that GM3 was the predominant ganglioside, followed by GM2, GM1, GD3 and some traces of GD1a in human epidermis [32]. We did not detect GD1a in our samples; however, our detection of GM3, GM2, GM1, and GD3 was similar to their results. We identified 23 species of HexCer and 15 species of LacCer, which are more than those previously reported by other groups [30,31]. Matrix-assisted laser desorption ionization–time of flight mass spectrometry analysis of cultured skin fibroblasts identified six species of HexCer [30]. Scherer et al. found four species of HexCer and LacCer (Hex_2_Cer) in primary human skin fibroblasts by hydrophilic interaction liquid chromatography–tandem mass spectrometry [31]. In our study, we identified 18 species of Gb3 and 15 species of Gb4. For both Gb3 and Gb4, d18:1/C24:0 and d18:1/C24:1, respectively, were the predominant species. Our results are the same as those reported for human dermal fibroblasts by Calvano et al. [27]. They reported that the most representative GSLs in skin fibroblasts were Hex_3_Cer (Gb3) followed by Gb4, HexCer, and Hex2Cer, while gangliosides were barely quantifiable. The amounts of the Gb3 species were in the order d18:1/C24:1 > d18:1/C24:0 > d18:1/C16:0 > d18:1/C22:0. Slight differences between our results and those of others, especially in the number of molecular species detected in each GSL, might be because of the type of column used. We previously showed that our method with a chiral column was able to separate neutral GSLs and gangliosides that cannot be separated by the commonly used C18 column [34]. Other reasons for the different results could be the passage and/or cell density of the fibroblasts. The senescence of skin fibroblasts is known to alter gene expression levels and to cause artifacts [35]. Vukelić and Kalanj-Bognar reported that the qualitative and quantitative composition of GSLs in cultured human skin fibroblasts changes with cell density [36].

As for the other major sphingolipid, SM, Valsecchi et al. identified 18 species in human fibroblasts, and they reported that the most represented species were d18:1/C16:0, d18:1/C24:1, and d18:1/C24:0 [37]. Previously, we also quantified SM species along with acyl-CoA and phospholipids, such as phosphatidylcholine and phosphatidylethanolamine species, in fibroblasts from X-ALD patients and control subjects [38]. The predominant SM species were SM34:1 > SM42:2 > SM42:1, which were predicted as d18:1/C16:0, d18:1/C24:1, and d18:1/C24:0, respectively. These results are consistent with the current analysis; however, SM species detected at trace levels were not in complete agreement between the current and previous analyses. Additionally, we previously found that SM 34:1 and SM 44:1 were present at significantly higher levels in CCALD fibroblasts compared with those in control fibroblasts, whereas SM44:1 and SM 44:2, identified as SM d18:1/C26:0 and d18:1/C26:1, were significantly increased in fibroblasts from CCALD patients compared with those from control subjects. These discrepancies might be caused by the small sample size, or the passage and/or cell density of the fibroblasts. Interestingly, this study, together with our previous study on SM species in X-ALD patient fibroblasts [38] indicate that GSLs contain higher amounts of C24 VLCFAs compared with shorter FAs such as C16, whereas SMs contain C16 more abundantly than C24. SM is produced by the transfer of a phosphocholine moiety to ceramide by sphingomyelin synthase (SMS) [39]. However, most GSLs are converted from ceramide by glucosylceramide synthase (GCS) [40]. Therefore, differences in the main fatty acid species of SMs and GSLs might be caused by the specificities of the two enzymes, SMS and GCS. For further analysis, it will be important to overcome some limitations of the method used in this study. Our current method has difficulty in distinguishing between glucosylceramide and galactosylceramide isomers in biological samples. Therefore, further refinement using an appropriate internal standard is needed to identify those two isomers in biological samples, which contain unknown GSL species. Another issue is that absolute amounts cannot be compared between the lipid classes because the electrospray ionization efficiency of lipids may vary with structure. Synthesizing specific internal standards for individual GSL species should be considered for future studies.

As mentioned in the introduction, there are no obvious correlations between phenotypes and genotypes in X-ALD patients, and the phenotypic variation of X-ALD is high. Thus, there is an urgent need to identify metabolite biomarkers to predict the phenotype and progression of X-ALD. Once X-ALD is suspected, two diagnostic tests can be performed. One is magnetic resonance imaging (MRI) to identify the degree of damage and progression of the disease. The other is a blood test that analyzes plasma VLCFA levels, such as C24:0, C25:0, and C26:0, which are elevated in ALD patients, and their ratio to C22:0 [17]. To improve patient prognosis, early and prompt diagnosis at the onset of the disease is critical. Here, we revealed that GSLs containing C25 and C26-VLCFA were elevated in fibroblasts from X-ALD patients. The *de novo* ceramide synthesis pathway begins with the conversion of serine and palmitoyl-CoA into dihydrosphingosine, then (dihydro)ceramide synthase (CerS) acylates dihydrosphingosine to yield dihydroceramide, followed by a desaturation to form ceramide [41]. Each isozyme of CerS may have a preference for acyl-CoAs with specific chain lengths and/or degree of saturation. We previously reported that acyl-CoA and phosphatidylcholine species with 24 and 26 carbons accumulate in fibroblasts from X-ALD patients, whereas SM with 26 carbons but not with 24 carbons accumulate in fibroblasts from X-ALD patients [17], indicating that C24-CoA increases in the absence of ABCD1, leading to the increase C26-CoA through a FA elongation pathway. C26-CoA is then efficiently converted to other metabolites, such as PLs. Our results in this study indicate that increased C26-CoA can be converted to GSLs as well as PLs and that C26-CoA accumulates in X-ALD patients.

As previously reported, VLCFA levels are elevated in male X-ALD patients, but the correlation of VLCFA levels with the severity or the onset of clinical phenotypes [2,42,43] remains unclear. Therefore, it is still not known whether the accumulation of VLCFAs in X-ALD itself causes demyelination or even contributes to the progression of X-ALD. Further analysis of VLCFA metabolism, including measuring GSL species and levels, will be useful to reveal the relevance of the onset or severity of the disease in different X-ALD subtypes.

## 4. Materials and Methods

### 4.1. Reagents

Internal standards, N-C16:0-CD_3_-glucosylceramide and N-C18:0-CD_3_-Gb3, were obtained from Matreya LLC (State College, PA, USA). N-C16:0-D_31_-D-erythro-sphingosylphosphorylcholine was obtained from Avanti Polar Lipids (Birmingham, AL, USA). All solvents of LC–MS grade were purchased from Wako Pure Chemical Industries (Osaka, Japan).

### 4.2. Cell Lines and Cell Culture

Primary human fibroblasts were established from skin biopsy samples of X-ALD patients ((CCALD) and (AMN)) and their fathers as control subjects using previously reported procedures [44]. After initial expansion, the cells were frozen for further experiments. Cells were reconstituted from liquid nitrogen storage and cultured in minimal essential medium (Merck, Darmstadt, Germany) supplemented with 10% fetal bovine serum (BioWest, Nuaillé, France) and 2 mM L-glutamine-100 U/mL penicillin/streptomycin solution (FUJIFILM Wako Pure Chemical Corporation, Osaka, Japan) in 5% CO_2_ at 37 °C. Each cell line was passaged up to eight times before use.

### 4.3. Sample Preparation

Cells were scraped from 100 mm culture dishes in phosphate-buffered saline (PBS) and collected by centrifugation at 200× *g* for 5 min. The cell pellets were washed twice with PBS and then homogenized in 1 mL methanol. N-C16:0-CD_3_-glucosylceramide, N-C18:0-CD_3_-Gb3, and/or N-C16:0-D_31_-D-erythro-sphingosylphosphorylcholine were added to the homogenate as an internal standard (IS). The homogenate was further mixed with a vortex mixer and placed in a sonication bath for 10 min each. The homogenate was then centrifuged at 2000× *g* for 15 min at room temperature. The supernatant was dried under nitrogen gas and re-solubilized in methanol. The protein concentration in each homogenate was determined using a BCA protein assay kit (Thermo Fisher Scientific, Waltham, MA, USA).

### 4.4. LC–MS/MS Analysis

Quantitative analysis was performed on a QTRAP 4500 (SCIEX, Framingham, MA, USA) with an electrospray ionization (ESI) interface connected to a Nexera HPLC system (Shimadzu Corp., Kyoto, Japan). HPLC settings and the MRM transition method used were described previously [34]. MRM transitions were constructed to cover LCB with 18 carbons, and glycosphingolipid fatty acids with 14–30 carbons and 0–2 double bonds. Specific methods to detect HexCer and SM are described in Table 2. MRM transition covers the total of 32–48 carbon atoms and 0–3 double bonds of LCB and fatty acyl moieties for HexCer and SM. MRM transitions and respective collision energies for all analytes are listed in Table 2. MS with a Turbo Spray interface was operated in the positive ion mode under the following optimized conditions: curtain gas (CUR), 20 arbitrary units (A.U.); ion spray voltage (IS), 5500 V; temperature (TEM), 200 °C; collision gas (CAD), 9 A.U.; ion nebulizer gas (GS1), 40 A.U.; auxiliary gas (GS2), 40 A.U.; declustering potential (DP), 60 V; entrance potential (EP), 10 V; and collision cell exit potential (CXP), 15 V. The dwell time and the cycle time were 5 ms and 5 s, respectively. Nitrogen gas was used as the nebulizer, curtain, and collision gas. Analyst software 1.62 (SCIEX) and MultiQuant Software 3.0 (SCIEX) were used for data acquisition and quantitative analysis of data, respectively.

### 4.5. Statistical Analysis

Statistical analysis was performed using GraphPad Prism software version 9.0 with one-way analysis of variance (ANOVA). Post hoc analysis with Tukey’s test was used to compare the data between groups. *p* values less than 0.05, 0.01, 0.001, and 0.0001 are indicated with one, two, three, and four asterisks, respectively.

## 5. Conclusions

We profiled the GSL species in fibroblasts from X-ALD patients using our recently developed LC–MS method that uses a chiral column in positive ion mode. We found that VLCFAs, such as C25- and C26-containing GSLs and SMs, were elevated in fibroblasts of X-ALD patients. Importantly, our results indicate that only HexCer 44:1, predicted as d18:1/C26:0, showed a significant difference in abundance between CCALD and AMN fibroblasts. Further analysis to evaluate the use of GSLs in the diagnosis of X-ALD is necessary.

## Figures and Tables

**Figure 1 ijms-22-08645-f001:**
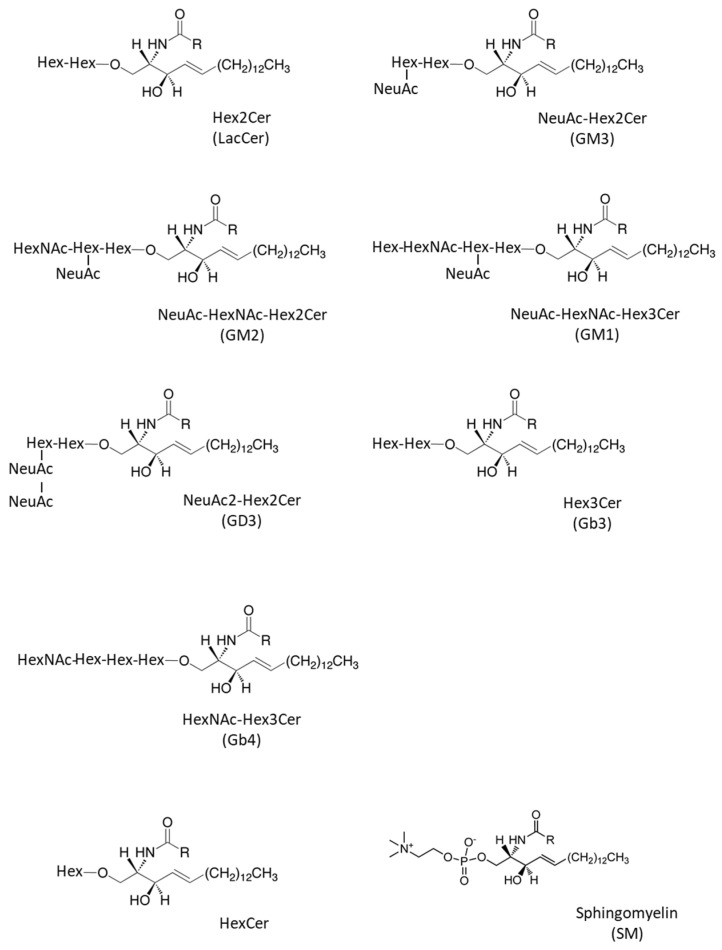
Structures of glycosphingolipids and sphingomyelin analyzed in this study. Hex; hexsosylceramide, HexNAc; N-acetylhexosamine, NeuAC; N-acetylneuraminic acid, and R; alkyl chain group.

**Figure 2 ijms-22-08645-f002:**
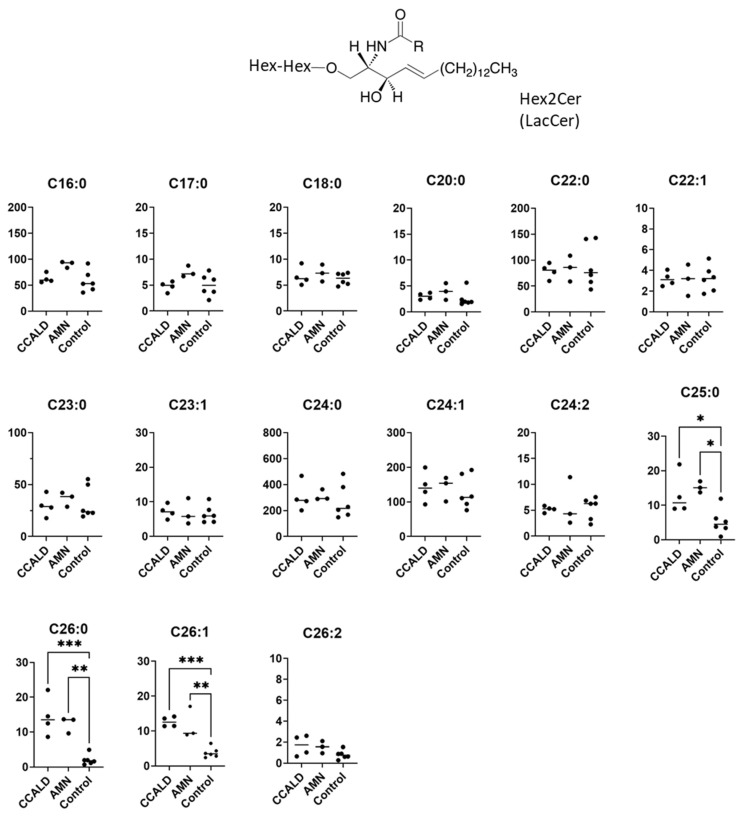
Amounts of Hex2Cer (LacCer) species in fibroblasts from X-ALD patients. The *Y*-axis of each graph shows the amount of each specific LacCer species as pmol/mg protein, calculated using the concentration of N-C18:0-CD_3_-Gb3 as an internal standard. Each species was quantified and classified according to the number of carbon atoms and double bonds, e.g., the C18:1 fatty acid species contains 17 carbon atoms and 1 double bond in the R (alkyl chain) group. * *p* < 0.05, ** *p* < 0.01, and *** *p* < 0.001.

**Figure 3 ijms-22-08645-f003:**
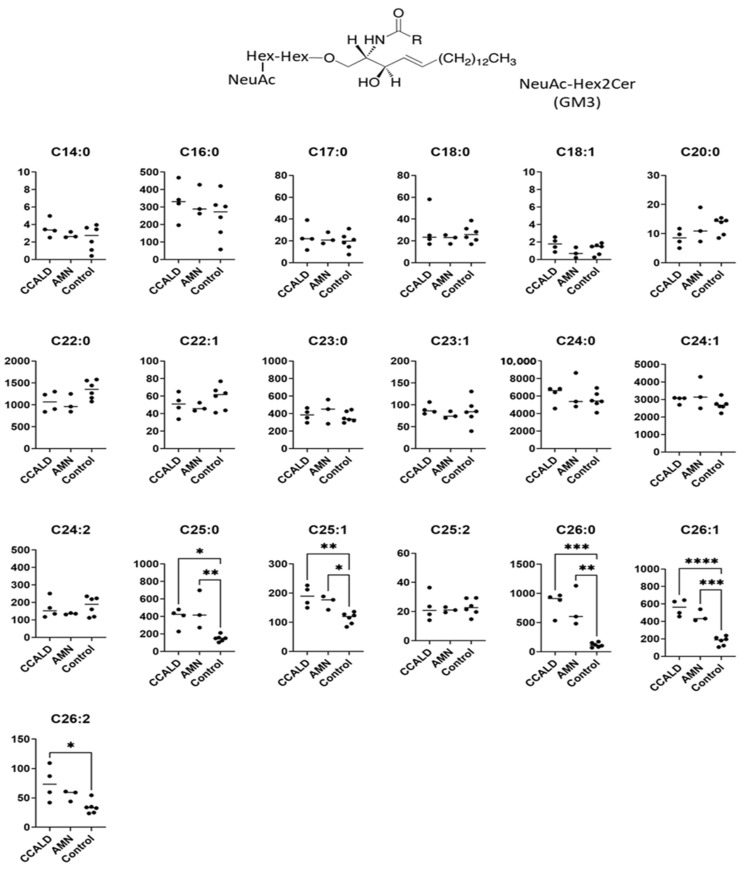
Amounts of NueAC-Hex2Cer (GM3) species in fibroblasts from X-ALD patients. The *Y*-axis of each graph shows the amount of each specific GM3 species as pmol/mg protein, calculated using the concentration of N-C18:0-CD_3_-Gb3 as an internal standard. Each species was quantified and classified according to the number of carbon atoms and double bonds, e.g., the C18:1 fatty acid species contains 17 carbon atoms and 1 double bond in the R (alkyl chain) group. * *p* < 0.05, ** *p* < 0.01, *** *p* < 0.001, and **** *p* < 0.0001.

**Figure 4 ijms-22-08645-f004:**
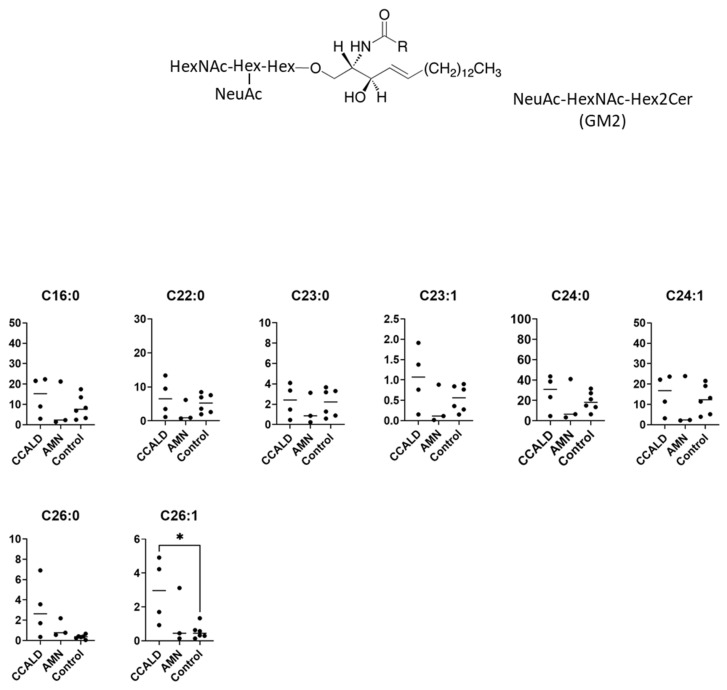
Amounts of NeuAc-HexNAc-Hex2Cer (GM2) species in fibroblasts from X-ALD patients. The *Y*-axis of each graph shows the amount of each specific GM2 species as pmol/mg protein, calculated using the concentration of N-C18:0-CD_3_-Gb3 as an internal standard. Each species was quantified and classified according to the number of carbon atoms and double bonds, e.g., the C18:1 fatty acid species contains 17 carbon atoms and 1 double bond in the R (alkyl chain) group. * *p* < 0.05.

**Figure 5 ijms-22-08645-f005:**
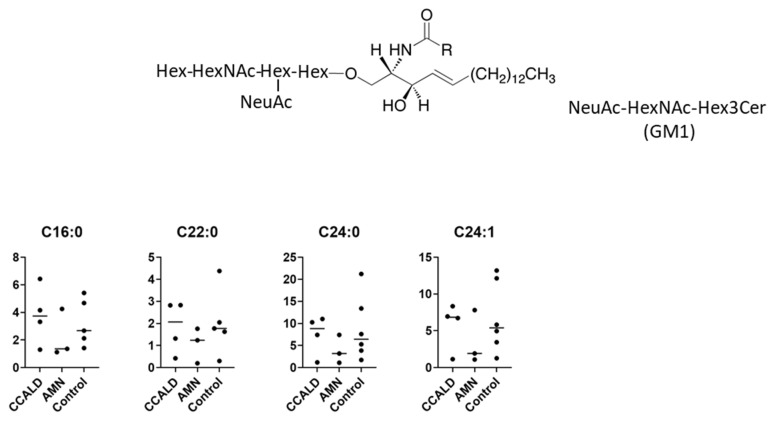
Amounts of NeuAc-HexNAc-Hex3Cer (GM1) species in fibroblasts from X-ALD patients. The *Y*-axis of each graph shows the amount of each specific GM1 species as pmol/mg protein, calculated using the concentration of N-C18:0-CD_3_-Gb3 as an internal standard. Each species was quantified and classified according to the number of carbon atoms and double bonds, e.g., the C16:1 fatty acid species contains 15 carbon atoms and 1 double bond in the R (alkyl chain) group.

**Figure 6 ijms-22-08645-f006:**
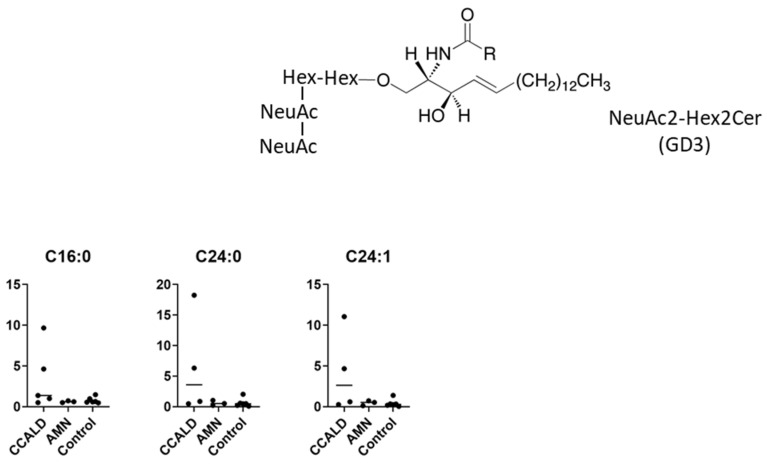
Amounts of NeuAc2-Hex2Cer (GD3) species in fibroblasts from X-ALD patients. The *Y*-axis of each graph shows the amount of each specific GD3 species as pmol/mg protein, calculated using the concentration of N-C18:0-CD_3_-Gb3 as an internal standard. Each species was quantified and classified according to the number of carbon atoms and double bonds, e.g., the C16:1 fatty acid species contains 15 carbon atoms and 1 double bond in the R (alkyl chain) group.

**Figure 7 ijms-22-08645-f007:**
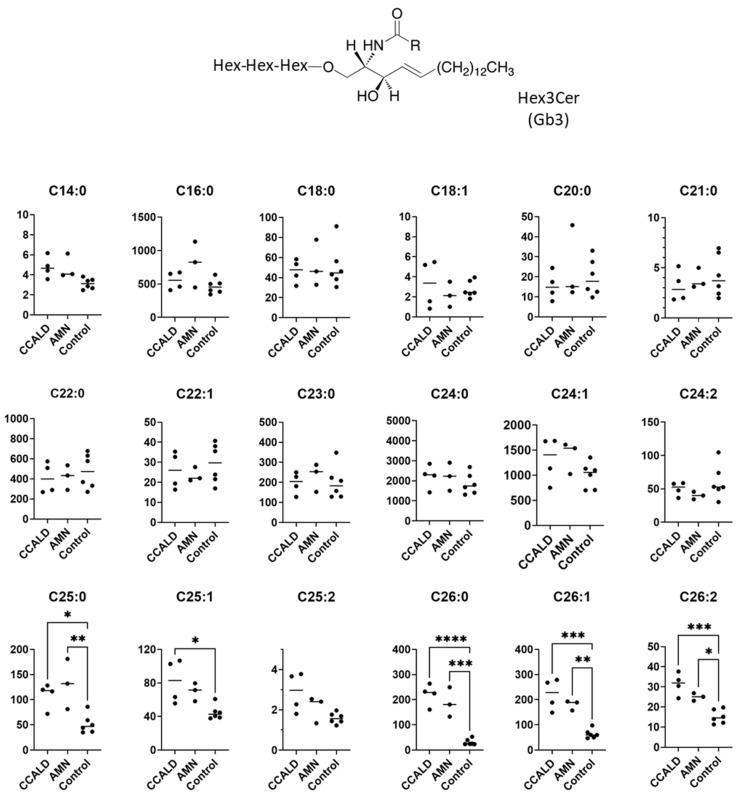
Amounts of Hex3Cer (Gb3) species in fibroblasts from X-ALD patients. The *Y*-axis of each graph shows the amount of each specific Gb3 species as pmol/mg protein, calculated using the concentration of N-C18:0-CD_3_-Gb3 as an internal standard. Each species was quantified and classified according to the number of carbon atoms and double bonds, e.g., the C18:1 fatty acid species contains 17 carbon atoms and 1 double bond in the R (alkyl chain) group. * *p* < 0.05, ** *p* < 0.01, *** *p* < 0.001, and **** *p* < 0.0001.

**Figure 8 ijms-22-08645-f008:**
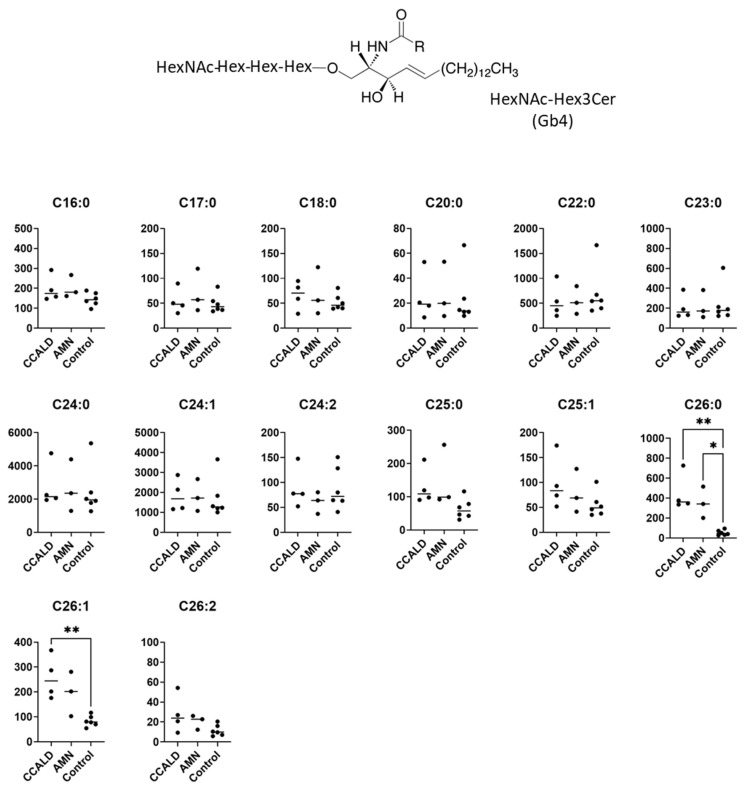
Amounts of HexNAc-Hex3Cer (Gb4) species in fibroblasts from X-ALD patients. The *Y*-axis of each graph shows the amount of each specific Gb4 species as pmol/mg protein, calculated using the concentration of N-C18:0-CD_3_-Gb3 as an internal standard. Each species was quantified and classified according to the number of carbon atoms and double bonds, e.g., the C18:1 fatty acid species contains 17 carbon atoms and 1 double bond in the R (alkyl chain) group. * *p* < 0.05 and ** *p* < 0.01.

**Figure 9 ijms-22-08645-f009:**
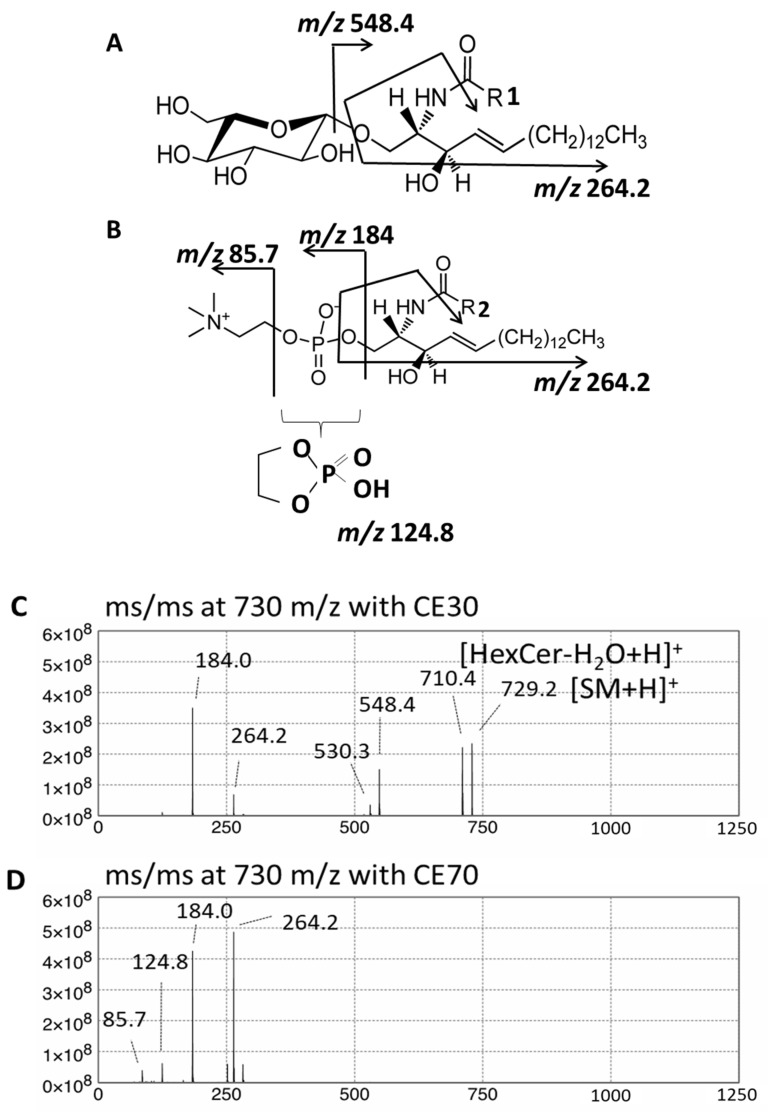
Fragmentation patterns of GalCer (**A**) and SM (**B**) in positive ion mode. Both GalCer and SM generate a product ion with 264 *m*/*z*. MRM traces for transitions from *m*/*z* 728.6 to 548.4 for GalCer d18:1/C18:0 and from *m*/*z* 729.6 to 184 for SM d18:1/C18:1. Assignment of LC–MS/MS spectra at 730 *m*/*z* for a mixture of GalCer d18:1/C18:0 and SM d18:1/C18:1 with collision energy (CE) of 30 V (**C**) and 70 V (**D**).

**Figure 10 ijms-22-08645-f010:**
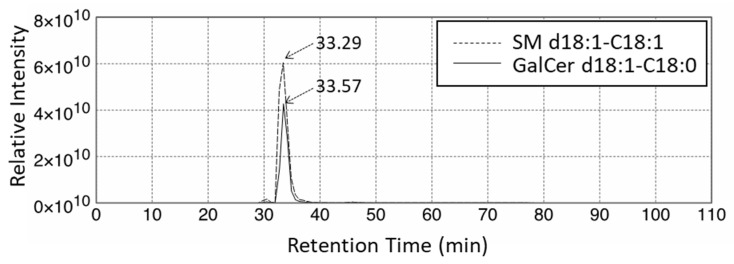
Retention times of GalCer d18:1/C18:0 and SM d18:1/C18:1.

**Figure 11 ijms-22-08645-f011:**
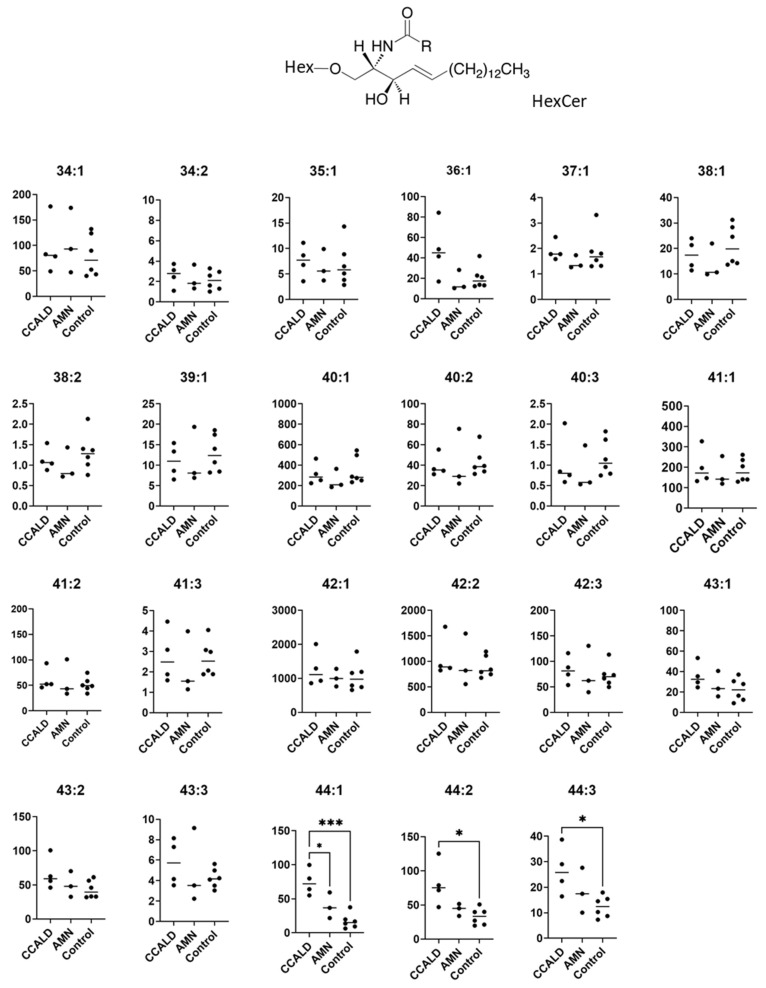
Amounts of HexCer species in fibroblasts from X-ALD patients. The *Y*-axis of each graph shows the amount of each specific HexCer species as pmol/mg protein, calculated using the concentration of N-C16:0-CD_3_-glucosylceramide as an internal standard. Each species was quantified and classified according to the total number of carbon atoms and double bonds in the ceramide moiety, e.g., the C34:1 species contains 33 carbon atoms and 1 double bond in the fatty acid and long-chain base moiety. * *p* < 0.05 and *** *p* < 0.001.

**Figure 12 ijms-22-08645-f012:**
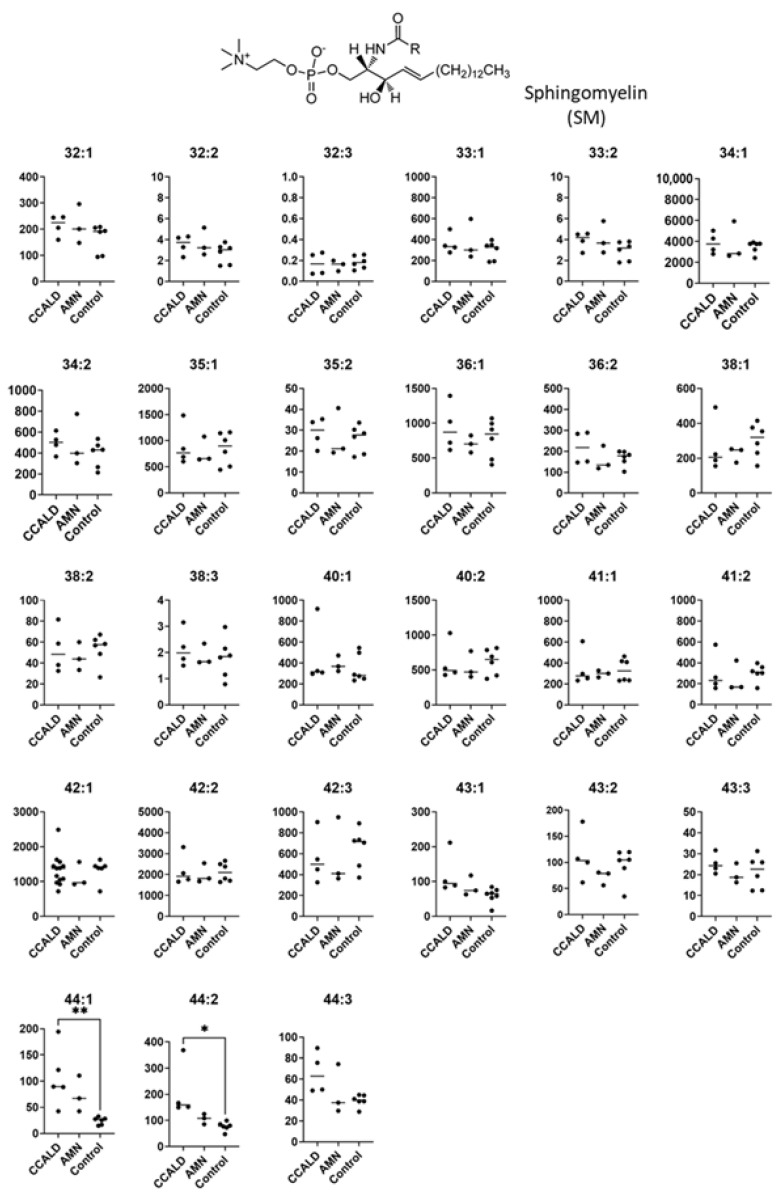
Amounts of sphingomyelin (SM) species in fibroblasts from X-ALD patients. The *Y*-axis of each graph shows the amounts of specific SM species as pmol/mg protein, calculated using the concentration of N-C16:0-D_31_-D-erythro-sphingosylphosphorylcholine as an internal standard. Each species was quantified and classified according to the total number of carbon atoms and double bonds in the ceramide moiety, e.g., the C34:1 species contains 33 carbon atoms and 1 double bond in the fatty acid and long-chain base moiety. * *p* < 0.05 and ** *p* < 0.01.

**Figure 13 ijms-22-08645-f013:**
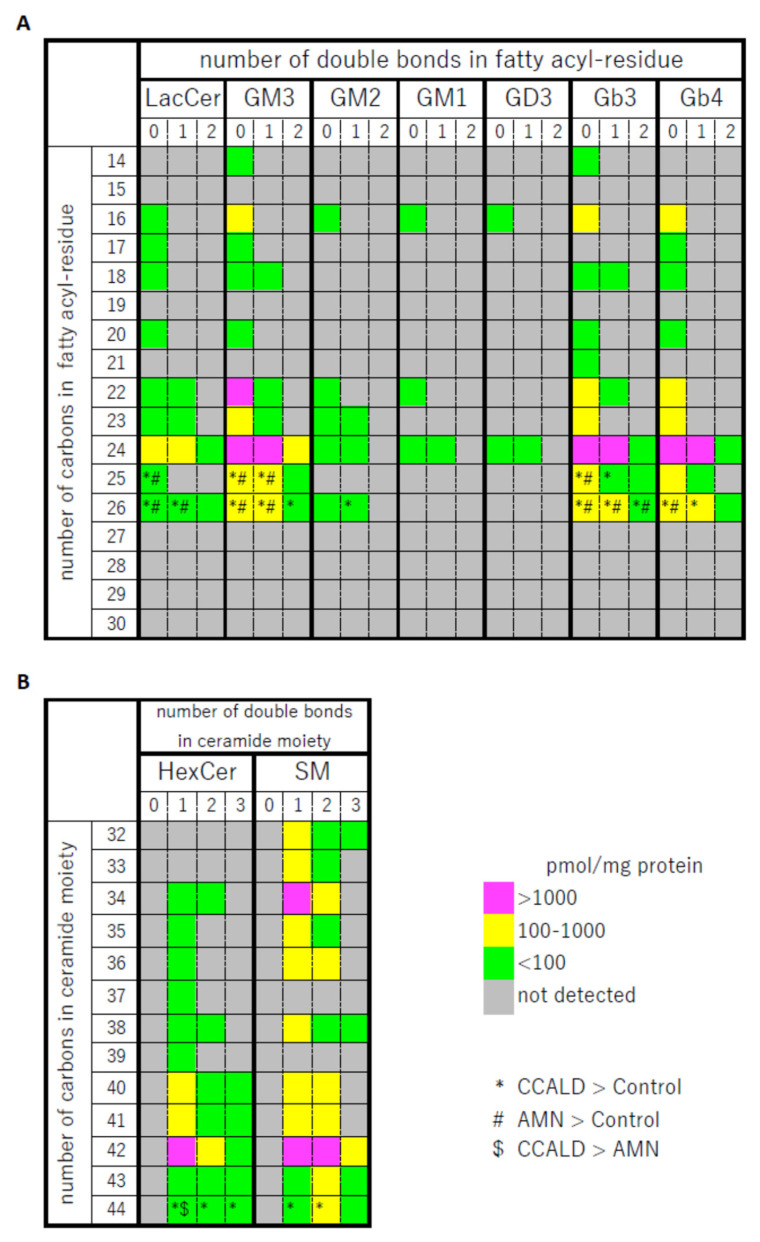
Quantity of each glycosphingolipid (GSL) and sphingomyelin (SM) species in X-ALD fibroblasts and comparison of the amounts of each GSL and SM species among groups. LacCer, GM3, GM2, GD3, GB3, and Gb4 are shown in (**A**), and HexCer and SM are shown in (**B**). The mean quantity of each species in fibroblasts from X-ALD patients is represented as pmol/mg protein with a color code. Each pmol/mg protein value was calculated using the concentration of N-C18:0-CD_3_-Gb3 for LacCer, GM3, GM2, GD3, GB3, and Gb4 (A), N-C16:0-CD_3_-glucosylceramide for HexCer, and N-C16:0-D_31_-D-erythro-sphingosylphosphorylcholine for SM (**B**) as an internal standard. Each GSL and SM species that was significantly more abundant in CCALD or AMN samples relative to control samples is indicated as * and #, respectively. Each GSL and SM species that was significantly more abundant in CCALD samples relative to AMN samples is indicated as $.

**Table 1 ijms-22-08645-t001:** Summary of the abbreviated name, systematic name, common name, and the references for each GSL.

Abbreviated Name	Systematic Name	Common Name	Ref
HexCer	Glc/Galβ-Cer	HexCer	[27,28,29,30,31]
Hex2Cer	Galβ1-4Glcβ-Cer	LacCer	[27,28,29,31]
NeuAc-Hex2Cer	NeuAcα2-3Galβ1-4Glcβ-Cer	GM3	[27,29,32]
NeuAc-HexNAc-Hex2Cer	GalNAcβ1-4(NeuAcα2-3)Galβ1-4Glcβ-Cer	GM2	[27,32]
NeuAc-HexNAc-Hex3Cer	Galβ1-3GalNAcβ1-4(NeuAcα2-3)Galβ1-4Glcβ-Cer	GM1	[27]
NeuAc2-Hex2Cer	NeuAcα2-8NeuAcα2-3Galβ1-4Glcβ-Cer	GD3	[28,29,32]
Hex3Cer	Galα1-4Galβ1-4Glcβ-Cer	Gb3	[27,28,29]
HexNAc-Hex3Cer	GalNAcβ1-3Galα1-4Galβ1-4Glcβ-Cer	Gb4	[27,29,33]

**Table 2 ijms-22-08645-t002:** MRM transitions and collision energies for LC–MS/MS analysis.

Analytes	MRM Transitions	CE (V)
LacCer	M + H → 264.2	500 *m*/*z*~; 70, 700 *m*/*z*~; 80, 1000 *m*/*z*~; 90
GM3		
GM2		
GM1		
GD3		
Gb3		
Gb4		
HexCer	Neutral loss of 180	500 *m*/*z*~; 30, 900 *m*/*z*~; 40, 1000 *m*/*z*~; 50
SM	M + H → 184	70

## Data Availability

Data that support the findings of this study are contained within the article are available on request.

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
