# Peer review of "Glycosphingolipids with Very Long-Chain Fatty Acids Accumulate in Fibroblasts from Adrenoleukodystrophy Patients"

_ijms, 2021, doi:10.3390/ijms22168645_

Round 1

Reviewer 1 Report

The paper is interesting and the research sound. The authors hint to a possible utility of MS of cultured fibroblast glycosphingolipids of X-ALD patients to distinguish between the two main phenotypes: CCALD and AMN. HexCer C26:0 might be differentially elevated in CCALD vs AMN. although more data is needed. They also confirm the elevated very-long fatty acid GSL in X-ALD patients already seen by other authors using MS albeit minor differences. I have to congratulate the authors for a very clear exposition and presentation of their research.

I have only four specific comments:

1: I would move the sentence “primary human fibroblasts were established from skin …. control subjects” from the Institutional Review Board Statement to the methods section. I would also add the methodology used to obtain skin biopsy, tissue disaggregation (which enzymes were used and procedures) and isolation of cells until the fibroblast culture is achieved. Were the primary fibroblasts expanded? How many passages? Were they frozen? That is important for both the results and for the sake of clarity. There is not even a reference. If it is very long it can be briefly stated in the Cell culture section and expanded in supplementary information. But it cannot be overlooked, it has to be explained.

2: About the supplementary information it is stated on lines 276-277 that “… specific methods to detect HexCer and SM are described in Supplemental Materials”. I was not able to find this information in the manuscript. For the sake of clarity, the Supplemental Materials: text, tables or figures should be organized in a separate section leaving them apart from the main article in the revised manuscript. It will facilitate future revisions and publication. Having figures S1 and S2 in the midst of the manuscript was confusing.

3: Figures 10 and 11 could be consolidated. Figure 11 information could be added into 10 as it uses graphical symbols that would be seen over the colored cells. Maybe a lighter shade of green and purple would be necessary. It would compact the information making it easier for the reader to have a general summary of the results which seems to be the purpose of these figures.

4: A very minor suggestion: the nomenclature of HexCer species could include the predicted form to be able to compare to the other GSLs. I understand the total carbon:total double bonds notation is useful to compare with SM but in order to be consistent with the other GSL a predicted fatty acid carbon:double bond notation should be included: 44:1 (C26:0) for example. Figures 8 and 10 should be modified correspondingly.

Reviewer 2 Report

Dear authors,

The article is well-written. I am pleased to tell you that your presentation of data is straightforward and accurate. Overall I believe the data and technologies are solid. 

I wish you could answer the following questions and read the following comments that may help you make the article even stronger. 

  1. Line 34 and line 233 write about the phenotypes not correlated with genotypes. Please be more specific on what phenotypes you refer to. I wonder if it is about severity of the symptoms.
  2. It would be clearer if you define VLCFA and other FA, for instance, LCFA. 
  3. Line35 mentions about phenotypes and distribution varying by countries, but be more specific, and it should have references. 
  4. In most lipids, C26 or C25 fatty acid content is increased but not C24, although C24-containing lipids are more abundant in many lipid species than C26 series. However, as far as I know, ABCD1 can utilize C24 and C26 equivalently. I would like to hear your opinions on why the amount of C26 lipids, not C24 lipids, increases. 
  5. Line228 emphasizes SM contains shorter fatty acids, but GSL also has shorter fatty acids quite abundantly. I do not understand the point of this part. Please make this part clear. 
  6. It would be difficult to include more samples at this stage, but I wonder if you can find the measurement about the severity of disease in your patient cells and if you can correlate it with VLCFA content. Also, it can be a good argument if you could treat cells with Lorenzo's oil or similar treatment and compare the lipidome with patients. 

Reviewer 3 Report

Fujiwara et al profiled the glycosphingolipid pools of X-linked ALD and found that the very long chain fatty acid containing GSLs were elevated in two affected patient populations compared to controls. While the results are not entirely novel, there are some potentially interesting insights from the data and the more complete profiling from the methodology is useful.

  1. The introduction is very brief, this should be expanded to include more information on the disease and the metabolites profiled and the role they play in the disease and more references included
  2. The manuscript would be quite challenging to read for non specialists due to the many abbreviations. I think adding the structures to table 1 next to the name and abbreviation may aid with this, as well as having them with the graphs.
  3. While I appreciate there are many different species that have been measured with the many different graphs, I wonder whether something such as a heatmap may be more applicable to present the data. That would enable a better comparison between the species abundances in addition to between disease states. If not, it would be useful to have a y-axis label.
  4. Considering the challenges with the measurements and the delineation between species, it may be useful to add a limitations section which highlights these issues.

Reviewer 4 Report

This is a full-length research report of profiling glycosphingolipid (GSL) spices in fibroblasts in patients with childhood cerebral X-linked adrenoleukodystrophy (CCALD) and adrenomyeloneuropathy (AMN), compared with control subjects, to identify functional biomarkers which diagnose phenotype of X-ALD. The authors have tried to reveal specific distribution of GSLs that might function biomarkers for certain X-ALD, suggesting that the profile of GSLs in fibroblasts in patents with X-ALD might act biomarker(s) for the close relationship between genotypes and phenotypes among X-ALD. This issue has progressively been focused on clinical stages. This paper is well structured, written, and discussed on their topic, and overall impact of their research seems to be considered strong. First, to enhance the strength of this paper, please describe what exact differences explain the relationship between genotypes and phenotypes in patients with X-ALD based on their work clearly in the abstract and the discussion section. The readers wish to know a take-home message regarding authors’ direction to better improving diagnostic aspect of X-ALD diseases. Second, please describe authors’ perspective why the different patterns of GSLs determine precise phenotypes of the diseases. This content is important part of the discussion of the revised manuscript.

Reviewer 5 Report

The manuscript is readable and understandable. I think it would be suitable for publication.

Figure S1 and Figure S2 should be moved in the supplementar material file.

Author Response

Thank you for your helpful  comments. 

We have rewritten the sentence to clarify the meaning and renumbered all supplemental figures as main figures. Figure S1 and Figure S2 are now Figure 9 and Figure 10, respectively.

Round 2

Reviewer 4 Report

The authors have addressed properly all the issues raised by reviewers including me.  I have no more comments, and now recommend that this manuscript is acceptable for publication in the IJMS.